# SIRI: Scaling Iterative Reinforcement Learning with Interleaved Compression

## Abstract

We introduce SIRI, **S**caling **I**terative **R**einforcement Learning with **I**nterleaved Compression, a simple yet effective RL approach for Large Reasoning Models (LRMs) that enables more efficient and accurate reasoning. Existing studies have observed repetitive thinking patterns in LRMs, and attempts to reduce them often come at the cost of performance. In this paper, we show that this trade-off can be overcome through a training regime that iteratively alternates between compressing and expanding the reasoning budget, by dynamically adjusting the maximum rollout length during training. The *compression phase* cuts the rollout length, forcing the model to make precise and valuable decisions within a limited context, which effectively reduces redundant tokens and increases reasoning density. The *expansion phase* then relaxes the length limit, providing space for the model to explore and plan in long-horizon settings. Remarkably, we find that after each compression–expansion cycle, the model's performance improves even as its output length decreases, steadily pushing it closer to the Pareto frontier in the performance–efficiency trade-off. Training on DeepSeek-R1-Distill-Qwen-1.5B, SIRI-low improves performance on AIME24 by 43.2% while reducing token usage by 46.9% after three iterations, and SIRI-high achieves the highest accuracy compared to all other methods (Figure 1). Our findings shed light on the potential of periodically oscillating the LRM's output truncation length during training to dynamically balance exploration and efficiency in reasoning, converging towards an optimal "sweet spot" between the two.

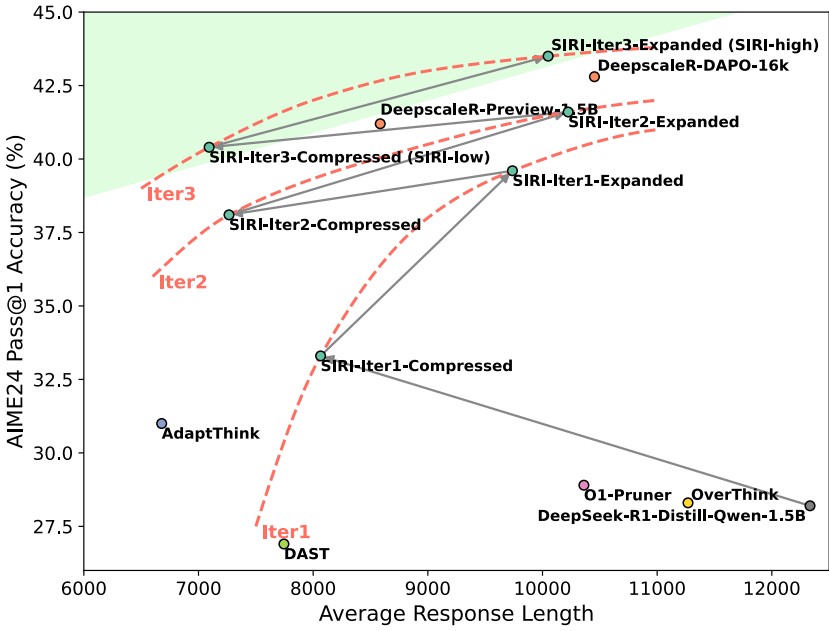

Figure 1: Performance-efficiency comparison between different training methods applied to DeepSeek-R1-Distill-Qwen-1.5B. SIRI continually pushes the model to the Pareto frontier.

## 1 INTRODUCTION

Large Language Models (LLMs) have become the frontier of AI research, demonstrating impressive performance across a wide range of domains, such as text generation, code generation, math reasoning, and autonomous agents (OpenAI, 2025). In particular, Large Reasoning Models (LRMs) (OpenAI, 2024; DeepSeek-AI, 2025; Zeng et al., 2025), a branch of LLMs tailored for reasoning tasks such as math, physics, and coding, have witnessed a great leap recently, empowered by large-scale reinforcement learning (RL) algorithms (Schulman et al., 2017; Shao et al., 2024). These models are trained using Test-Time Scaling strategy (Snell et al., 2024; Muennighoff et al., 2025) that appears as a long Chain-of-Thought (Wei et al., 2022), utilizing the model's backtracking, verification, exploration and iterative refinement abilities at test time to obtain superior reasoning capability.

However, although RL can boost the model's performance on reasoning tasks, it also inevitably causes a rapid increase in the model's output length. The amount of useless reasoning and overthinking by the model rises significantly (Chen et al., 2024a; Qu et al., 2025), leading to a notable increase in both training and inference time. To overcome this issue, prior works have attempted approaches such as fine-tuning with short and precise reasoning traces (Kang et al., 2025; Ma et al., 2025), introducing length penalties or length truncation during RL (Team et al., 2025; Aggarwal & Welleck, 2025; Luo et al., 2025a), and adopting hybrid reasoning strategies to automatically switch between thinking and non-thinking (Fang et al., 2025; Zhang et al., 2025; Lou et al., 2025) to improve the token efficiency of LRMs. However, without exception, these methods degrade the model's performance or cause it to stagnate, preventing it from fully unlocking its capabilities. To support this, in Figure 1, all other length-compression approaches perform worse than models trained with standard RL (DeepScaleR) by a large margin, placing them inside the Pareto frontier.

In this paper, we propose SIRI, **S**caling **I**terative **R**einforcement Learning with **I**nterleaved Compression, a simple yet effective framework that pushes the Pareto frontier by simultaneously reducing token usage and improving reasoning accuracy. The key idea of SIRI is to **periodically alternate between compressing and expanding the reasoning budget during training**, by dynamically adjusting the maximum rollout length according to a cosine scheduler. The compression phase forces the model to think concisely by reducing overthinking, while the expansion phase encourages the model to further explore based on mature reasoning traces. Unlike prior approaches that suffer from a strict efficiency-performance trade-off, SIRI leverages the compression–expansion cycle to achieve steady gains in accuracy despite shorter outputs. As shown in SIRI's evolution trace in Figure 1, with each iteration the model spirals upward by using fewer tokens and achieving higher performance. We contribute the core factor in SIRI's success to compressing length in each iteration while not letting accuracy fall off the cliff. We further find that SIRI generalizes effectively across different model sizes.

Empirically, our method demonstrates superior performance against state-of-the-art models trained under the same 16K output limit. Training based on DeepSeek-R1-Distill-Qwen-1.5B model, the expanded-length variant (SIRI-high) achieves a **43.5%** pass@1 on AIME24 with an average of **10K** tokens, while the compressed-length variant (SIRI-low) achieves **40.4%** pass@1 using only **7K** tokens on average. This nearly halves the token usage while still delivering a **43%** improvement over the initial pre-RL model.

## 2 RELATED WORK

### 2.1 LENGTH COMPRESSION FOR LRMs

Large reasoning models leverage test-time scaling to boost performance, but they frequently expend unnecessary tokens on repeated backtracking, unnecessary exploration, and non-reasoning filler (Hou et al., 2025). To mitigate such inefficiency, previous studies mostly leverage reward shaping for online RL training (Team et al., 2025; Aggarwal & Welleck, 2025; Luo et al., 2025b; Hou et al., 2025; Zhang et al., 2025) or precise reasoning traces for offline fine-tuning (Ma et al., 2025; Yang et al., 2025). The most commonly used reward shaping strategy is adaptive length penalty (Team et al., 2025), which is further augmented by prompt engineering (Aggarwal & Welleck, 2025). Another line of work apply budget forcing on models by either setting the reward to zero or forcing the model to end thinking and generate the final solution once the output length surpasses the token budget (Luo et al., 2025b; Hou et al., 2025; Muennighoff et al., 2025).

Recent studies also try to abandon the thinking process of relatively simple problems by training the model to generate end-of-thinking tokens (Yang et al., 2025; Zhang et al., 2025; Fang et al., 2025; Lou et al., 2025). However, all methods above inevitably compromise model performance compared to full-length RL training, while our work shows that we can close this gap by an iterative RL framework.

## 2.2 ITERATIVE TRAINING

Iterative training methods have been widely used in preference alignment and simple reasoning tasks because of their ability to leverage additional data generated by the model over iterations. One line of work uses maximum likelihood iterative training to enhance the model's reasoning abilities (Gulcehre et al., 2023; Singh et al., 2023). During each iteration, a dataset is first generated by the current model and labeled by a reward model. Then, this dataset is used to fine-tune the base model, yielding a stronger version for the next iteration. Another line utilizes positive and negative samples with Direct Preference Optimization (DPO) algorithms (Rafailov et al., 2023; Xiong et al., 2023; Chen et al., 2024b). In GSHF (Xiong et al., 2023), new chosen/rejected responses are generated in each iteration and added to the dataset for wider dataset coverage. In SPIN (Chen et al., 2024b), the chosen responses are generated by humans and fixed, while the model iteratively generates rejected responses and aligns with the human dataset. The most recent work unifies these two directions (Pang et al., 2024), applying both NLL and DPO loss for arithmetic tasks, witnessing modest gains in the GSM8K dataset. However, all methods above demonstrate only how iterative training can be applied in off-policy training scenarios for simple reasoning problems. In this work, we further show that iterative on-policy RL can be used to effectively advance the performance-efficiency Pareto frontier of LRMs.

## 3 PRELIMINARIES

### 3.1 PROBLEM FORMULATION

We consider a Large Language Model (LLM) parameterized by $\theta$, denoted by $\pi_\theta$ and a math dataset $\mathcal{D}$ with question-answer pairs, each denoted by $(\boldsymbol{x}, a^*)$. The model samples a problem $\boldsymbol{x}$ and generates a response $\boldsymbol{y} = [y_1, ..., y_m]$ sampled from the conditional distribution $\pi_\theta(\cdot|x)$. In the LLM's setting, each element in $\boldsymbol{y}$ is known as an output token. Specifically, for a Large Reasoning Model (LRM) trained on mathematical tasks with a fixed answer, the last output token, $y_m$, is the model's predicted answer for the problem. By defining the scoring function $R(\boldsymbol{y})$ and setting $R(\boldsymbol{y}) = 1$ if $a^* = y_m$ and $R(\boldsymbol{y}) = 0$ otherwise, we aim to find $\theta^*$ that satisfies

$$\theta^* = \arg\max_\theta \mathbb{E}_{(\boldsymbol{x}, a^*) \sim \mathcal{D}, \boldsymbol{y} \sim \pi_\theta(\cdot|\boldsymbol{x})} \Big[ R(\boldsymbol{y}) \mid |\boldsymbol{y}| \leq L \Big],$$

where $|\boldsymbol{y}|$ is the length of the output $\boldsymbol{y}$, and $L$ is the token budget.

### 3.2 GROUP RELATIVE POLICY OPTIMIZATION

Group Relative Policy Optimization (GRPO) (Shao et al., 2024), based on Proximal Policy Optimization (PPO) (Schulman et al., 2017), is widely used in post-training of LRMs. For each question-answer pair $(\boldsymbol{x}, a^*)$ sampled from dataset $\mathcal{D}_{\text{GRPO}}$, $\pi_\theta$ samples $G$ individual responses $\{\boldsymbol{y}_i\}_{i=1}^G$ and estimates the advantage of the $i$-th response with group-level rewards

$$\hat{A}_{i,t} = \frac{R(\boldsymbol{y}_i) - \text{mean}\big(\{R(\boldsymbol{y}_i)\}_{i=1}^G\big)}{\text{std}\big(\{R(\boldsymbol{y}_i)\}_{i=1}^G\big)}.$$

Then, the loss of the policy is calculated by

$$\mathcal{L}_{\text{GRPO}}(\theta) = -\mathbb{E}_{(\boldsymbol{x}, a^*) \sim \mathcal{D}_{\text{GRPO}}, \{\boldsymbol{y}_i\}_{i=1}^G \sim \pi_{\theta_{\text{old}}}(\cdot|\boldsymbol{x})}$$

$$\left[ \frac{1}{G} \sum_{i=1}^G \frac{1}{|\boldsymbol{y}_i|} \sum_{t=1}^{|\boldsymbol{y}_i|} \left( \min\Big( r_{i,t}(\theta) \hat{A}_{i,t}, \text{clip}\big(r_{i,t}(\theta), 1 - \epsilon, 1 + \epsilon\big) \hat{A}_{i,t} \Big) - \beta D_{\text{KL}}(\pi_\theta \| \pi_{\text{ref}}) \right) \right],$$

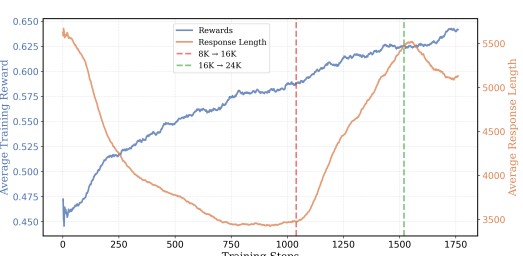 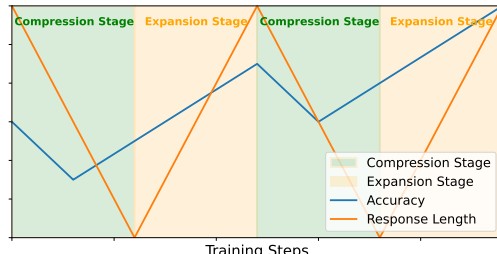

(a) DeepScaleR's training dynamics (Luo et al., 2025b).   (b) Hypothesized iteration dynamics of SIRI.

Figure 2: Motivation of SIRI: The compression stage primarily reduces the model's overthinking while preserving performance, storing potential to provide more room for exploration in the next interleaved expansion stage, and this process repeats cyclically.

where

$$r_{i,t}(\theta) = \frac{\pi_\theta(\boldsymbol{y}_{i,t} \mid \boldsymbol{x}, \boldsymbol{y}_{i,<t})}{\pi_{\theta_{\text{old}}}(\boldsymbol{y}_{i,t} \mid \boldsymbol{x}, \boldsymbol{y}_{i,<t})}.$$

In Dynamic Sampling Policy Optimization (DAPO) (Yu et al., 2025), the upper and lower clip thresholds are decoupled, and the former is set larger to encourage model exploration. Moreover, the KL divergence is removed in light that post-trained reasoning model will naturally diverge from the base model. We adopt these improvements in this work.

## 4 SIRI: SCALING ITERATIVE REINFORCEMENT LEARNING WITH INTERLEAVED COMPRESSION

### 4.1 MOTIVATION

In DeepScaleR's (Luo et al., 2025b) 8K training stage, there is an increase in the model's performance despite a sharp response length drop. This shows that *the model can compress key reasoning steps into shorter contexts, thus freeing capacity for exploration in the subsequent 16K stage*. However, the following context-expansion stage may again introduce redundant reasoning patterns. As illustrated in Figure 2, we hypothesize that interleaving compression with expansion can yield performance gains while maintaining comparable response lengths across expansion stages. The key ingredient of the success may lie in the compression stage: after the initial performance drop caused by switching from long to short outputs, it must restore performance to ensure the model does not fall below its level at the start of the next expansion stage. With this motivation, we now explore the best design for the compression-expansion schedule in the following subsections.

### 4.2 REWARD SHAPING

A common approach for length compression is reward shaping. We adopt the length-capping reward introduced in DeepScaleR, which assigns a reward to each response $\boldsymbol{y}$ based on a maximum length $L$ as follows:

$$R(\boldsymbol{y}) = \begin{cases} 1, & \text{if an answer can be extracted from clip}(\boldsymbol{y}, L) \text{ and is correct,} \\ 0, & \text{otherwise.} \end{cases}$$

Note that this method is effective when the responses in each group are diverse enough, i.e., there is a correct response whose length is lower than the capping threshold, and a correct/incorrect response whose length is higher than the capping threshold. In such case, the policy update will pose positive gradients on the short and correct responses, while posing negative gradients on the longer responses, directing the model to preserve correct and dense reasoning patterns while pruning inefficient or wrong patterns. On the other hand, while using an adaptive length penalty is mathematically

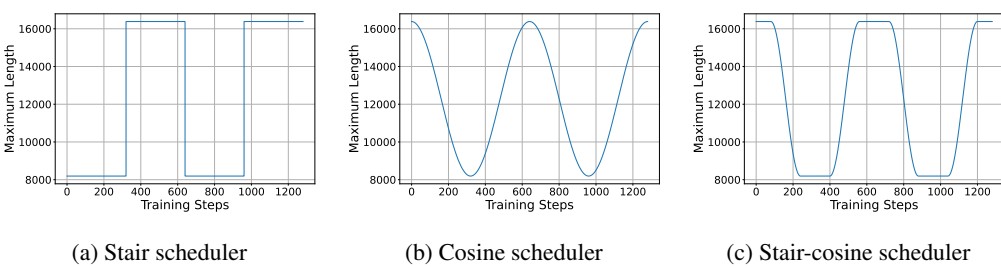

(a) Stair scheduler        (b) Cosine scheduler        (c) Stair-cosine scheduler

Figure 3: Illustration of different schedulers with cycle length 640.

justified, it requires complex hyperparameter tuning since different length constraints may have different optimal penalty coefficients. Moreover, their training efficiency is worse than the direct capping method. For these reasons, we do not adopt them.

### 4.3 LENGTH SCHEDULER

In iterative training, the design of the length scheduler is important as it controls the compression and exploration behavior of the model. The scheduler should have the following properties: **1)** prevent performance degradation during the compression phase, and **2)** encourage exploration during the expansion phase, meaning that the model's generation length should plateau before the expansion phase ends. Here, we introduce three types of schedulers. To unify notation, let $T$ denote the cycle length (in steps), $t$ be the current step, $L_{\max}$ and $L_{\min}$ denote the maximum and minimum capping threshold during each cycle. Figure 3 illustrates the curves of the respective schedulers.

**Stair scheduler**. The stair scheduler reduces the maximum generation length from the upper capping threshold $L_{\max}$ to the lower capping threshold $L_{\min}$ during the compression phase. It then switches from $L_{\min}$ to $L_{\max}$ when the model enters the expansion phase.

**Cosine scheduler**. To make the length reduction and recovery process smooth, we also investigate the cosine scheduler. The maximum generation length at each step $t$ can be written as

$$L = \frac{L_{\max} + L_{\min}}{2} + \frac{L_{\max} - L_{\min}}{2} \cdot \cos(\frac{2\pi}{T} \cdot t).$$

**Stair-cosine scheduler**. The cosine scheduler doesn't maintain at $L_{\max}$ and $L_{\min}$. However, this may hinder the model's ability to further explore at $L_{\max}$ after expansion and restore performance at $L_{\min}$ after compression. Thus, we combine the stair and cosine scheduler into a unified scheduler that ensures both smoothness of the whole process, exploration at $L_{\max}$, and exploitation at $L_{\min}$. Letting the current phase be $\phi = 2\pi \cdot \frac{t \bmod T}{T}$, the whole schedule can be written as

$$L = \begin{cases} L_{\max}, & \phi < \frac{\pi}{4} \ or \ \phi \geq \frac{7\pi}{4}, \\[2ex] \frac{L_{\max} + L_{\min}}{2} + \frac{L_{\max} - L_{\min}}{2} \cdot \cos\big(2(\phi - \frac{\pi}{4})\big), & \frac{\pi}{4} \leq \phi < \frac{3\pi}{4}, \\[2ex] L_{\min}, & \frac{3\pi}{4} \leq \phi < \frac{5\pi}{4}, \\[2ex] \frac{L_{\max} + L_{\min}}{2} + \frac{L_{\max} - L_{\min}}{2} \cdot \cos\big(2(\phi - \frac{3\pi}{4})\big), & \frac{5\pi}{4} \leq \phi < \frac{7\pi}{4}. \end{cases}$$

## 5 EXPERIMENTS

In this section, we conduct extensive experiments to validate our compression-expansion approach. Specifically, our experiments are designed to answer the following questions:

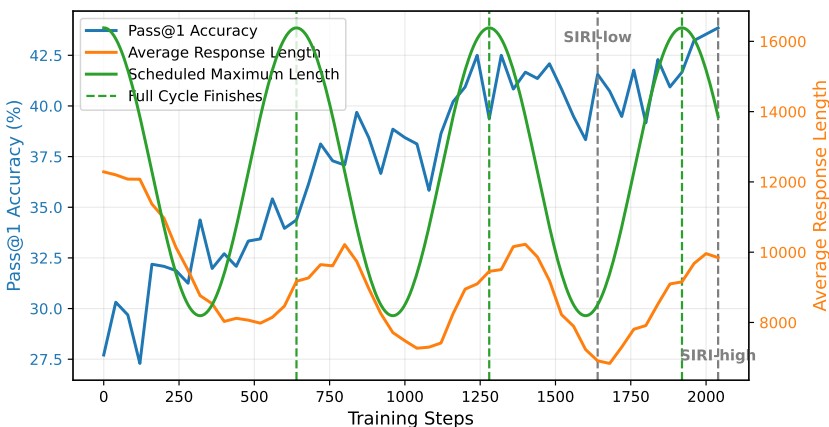

Figure 4: The 1.5B model's Pass@1 accuracy and average response length of SIRI with 640-cycle length cosine scheduler over three iterations on the AIME24 benchmark.

**RQ1**: Can the compression-expansion scheme enhance reasoning accuracy while pruning redundant tokens? What is the underlying mechanism behind this behavior?
**RQ2**: What is the best generalizable design of the length scheduler?
**RQ3**: Is the compression-expansion scheme generally applicable to different models?

## 5.1 EXPERIMENT SETUP

**Dataset**. To provide a fair comparison with the strong DeepScaleR (Luo et al., 2025b) baseline, we use the same training set used in training DeepScaleR-1.5B-Preview, which comprises 40K high-quality math questions with groundtruth answers selected from AIME 1983-2023, AMC, Omni-Math (Gao et al., 2024), and STILL (Min et al., 2024) datasets.

**Model**. For the initial pre-RL model, we select two representative open-source large reasoning models with different sizes: DeepSeek-R1-Distill-Qwen-1.5B and DeepSeek-R1-Distill-Qwen-7B (DeepSeek-AI, 2025), both fine-tuned on expert trajectories generated by DeepSeek-R1. For baseline comparison, we evaluate several popular RL approaches, including DeepScaleR-Preview (Luo et al., 2025b) (released checkpoint from the original DeepScaleR work), DAPO-DeepScaleR-16K (trained with DeepScaleR's 8K compression followed by 16K expansion schedule, but using DAPO's clip-higher and no KL-loss strategies for a fairer comparison with our method), OverThink (Chen et al., 2024a), DAST (Shen et al., 2025), O1-Pruner (Luo et al., 2025a), and AdaptThink (Zhang et al., 2025). **All baseline models are trained on the same dataset as ours.**

**Implementation Detail**. For RL training, we use the VeRL framework (Sheng et al., 2024). We adopt the GRPO (Shao et al., 2024) algorithm for training, but decouple the upper and lower thresholds for clipping, as well as removing the KL divergence, as proposed in DAPO (Yu et al., 2025). Specifically, we set 0.28 for clip-high and 0.2 for clip-low. For the length scheduler, we set $L_{max}$ at 16384 and $L_{min}$ at 8192. The models are trained with a sampling temperature of 1.0, a batch size of 128, and a learning rate of 1e-6. We use 8×H100 GPUs for training the 1.5B model and 16×H100 GPUs for the 7B model.

**Evaluation Configuration**. All the trained models are evaluated on AIME24, AIME25, AMC, and MATH500 (Hendrycks et al., 2021) datasets. We set the maximum generation length (including thinking tokens and answer tokens) at 16384, aligned with $L_{max}$ during training. We sample 32 outputs for each question during training, and sample 64 outputs for each question to obtain the final evaluation results shown in Table 1. The sampling temperature is set to 0.6. We report both the Pass@1 accuracy and the average token number of the responses.

## 5.2 RESULTS

Table 1 shows the main evaluation results. Our models, SIRI-low (SIRI-Iter3-Compressed) and SIRI-high (SIRI-Iter3-Expanded), were trained with a 640-cycle cosine length scheduler over three

Table 1: Performance comparison on AIME24, AIME25, MATH500 and AMC. Best result in **bold** and second best underlined.

| Method | AIME24 | | AIME25 | | AMC | | MATH500 | | Average |
|---|---|---|---|---|---|---|---|---|---|
| | Acc | Length | Acc | Length | Acc | Length | Acc | Length | $\frac{\Delta Acc}{CR}(\uparrow)^\star$ |
| *DeepSeek-R1-Distill-Qwen-1.5B* | | | | | | | | | |
| *Original* | 28.2 | 12333 | 21.5 | 12264 | 61.8 | 8449 | 82.4 | 4745 | 0.00 |
| DeepScaleR-Preview (Luo et al., 2025b) | 41.1 | 8585 | 29.0 | 8348 | 73.9 | 5515 | 87.6 | 3054 | 0.39 |
| DAPO-DeepScaleR-16K | 42.8 | 10453 | 30.9 | 10352 | 74.6 | 7339 | 88.1 | 4223 | 0.36 |
| OverThink (Chen et al., 2024a)[†] | 28.3 | 11269 | – | – | – | – | 81.2 | 4131 | 0.00 |
| DAST (Shen et al., 2025)[†] | 26.9 | 7745 | – | – | – | – | 83.0 | 2428 | 0.01 |
| O1-Pruner (Luo et al., 2025a)[†] | 28.9 | 10361 | – | – | – | – | 82.2 | 3212 | 0.01 |
| AdaptThink (Zhang et al., 2025) | 31.0 | **6679** | 22.3 | 6800 | 63.3 | **3498** | 82.0 | **1782** | 0.08 |
| SIRI-low (Ours) | 40.4 | 7093 | 29.6 | **6509** | 74.6 | 4700 | 87.7 | 2881 | **0.47** |
| SIRI-high (Ours) | **43.6** | 10049 | **32.2** | 9739 | **75.9** | 7396 | **88.4** | 4633 | 0.38 |
| *DeepSeek-R1-Distill-Qwen-7B* | | | | | | | | | |
| *Original* | 53.5 | 10306 | 38.3 | 11114 | 79.4 | 6740 | 90.2 | 3674 | 0.00 |
| DAPO-DeepScaleR-16K | **57.6** | 9983 | 40.8 | 10705 | 84.5 | 6508 | 92.5 | 3658 | 0.06 |
| OverThink (Chen et al., 2024a)[†] | 53.1 | 8744 | – | – | – | – | 89.4 | 2435 | 0.00 |
| DAST (Shen et al., 2025)[†] | 45.6 | 7578 | – | – | – | – | 89.6 | 2162 | 0.00 |
| O1-Pruner (Luo et al., 2025a)[†] | 49.2 | 9719 | – | – | – | – | 86.6 | 2534 | 0.00 |
| AdaptThink (Zhang et al., 2025) | 55.6 | 8546 | 37.0 | 9556 | 80.1 | 4778 | 90.6 | **1868** | 0.02 |
| SIRI-low (Ours) | 56.1 | **6122** | 41.5 | **6386** | 85.8 | 4015 | 93.5 | 2452 | 0.10 |
| SIRI-high (Ours) | 57.1 | 8585 | **45.4** | 9106 | **86.7** | 5773 | **93.7** | 3378 | **0.11** |

$^\star$ $\Delta Acc = \max(\frac{\text{current model accuracy}}{\text{initial model accuracy}} - 1, 0)$, CR (Compressed Ratio) $= \frac{\text{current model length}}{\text{initial model length}}$. Higher is better.

[†] For these methods, we directly use the results reported in AdaptThink (Zhang et al., 2025). Since the corresponding checkpoints were not released, we are unable to evaluate them on AIME25 and AMC.

iterations. Compared to the original 1.5B model, SIRI-low reduces response length by 43.1% and boosts performance by 27.0% on average. After expansion of SIRI-low during the third iteration, we yield SIRI-high that achieves the highest accuracy on all benchmarks, improving performance by 33.6% on average. A similar trend can also be seen on the 7B model. These show that the interleaved compression phase enhances, instead of mitigates, the model's potential to explore and plan in long Chain-of-Thought. Regarding generation length, while SIRI-low produces longer responses than models trained with adaptive length penalties (e.g., DAST) or "no-thinking" methods (e.g., Adapt-Think) on easier benchmarks (AMC and MATH500), its output length is comparable to them for the 1.5B model and notably shorter for the 7B model on more challenging benchmarks (AIME24 and AIME25). In addition, SIRI-low also performs similarly with DeepScalerR-Preview-1.5B (the latter is trained under 24K context). This demonstrates SIRI's robustness across tasks and its advantage on difficult problems.

We additionally report the accuracy-CR ratio that evaluates the change in the model's token efficiency after training. We find that SIRI trained models have the optimal accuracy-CR ratio, showing that iterative compression with a length scheduler is better at pruning redundant tokens compared to manually introducing "thinking" and "no-thinking" patterns (Zhang et al., 2025), or using complicated reward shaping techniques (Luo et al., 2025a). We detail our findings below.

**Token efficiency iteratively improves**. Figure 4 shows the training dynamic of the 1.5B model trained by the 640-cycle cosine scheduler. The model starts with an average response length of about 12000 tokens. After the first iteration, the average length is suppressed to about 8000 tokens with a 7% gain in accuracy. In the following iterations, we witness a stable increase in token efficiency, where the model ends each cycle with almost the same response length, but its Pass@1 accuracy consistently improves, eventually surpassing 43%. This shows that the model's reasoning is condensed through the iteration of compression and expansion. We also observe an interesting phenomenon: the change in model output length lags behind the scheduler. Typically, the output length reaches its peak or trough about 100-200 steps after the scheduled maximum or minimum length. This indicates that, thanks to the smoothness of the cosine scheduler, the model still has sufficient time to continue expanding or compressing its output length, even though the scheduler does not pause at the maximum or minimum length. Meanwhile, as shown in Figure 1, the model

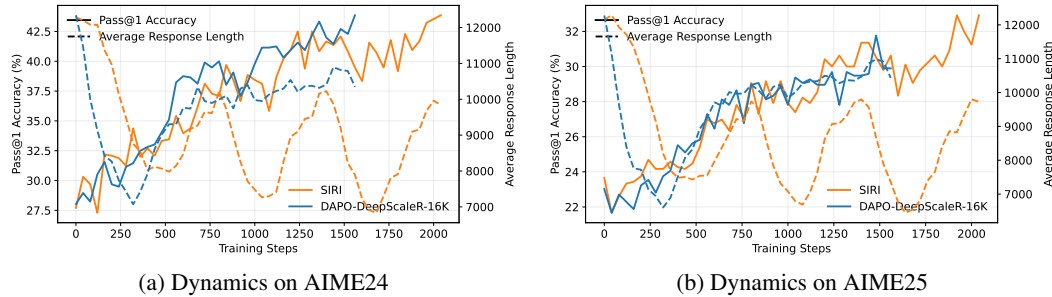

(a) Dynamics on AIME24          (b) Dynamics on AIME25

Figure 5: The training dynamics comparison between SIRI and DAPO-DeepScaleR-16K on DeepSeek-R1-Distill-Qwen-1.5B. DAPO-DeepScaleR-16K transits from 8K to 16K at step 320.

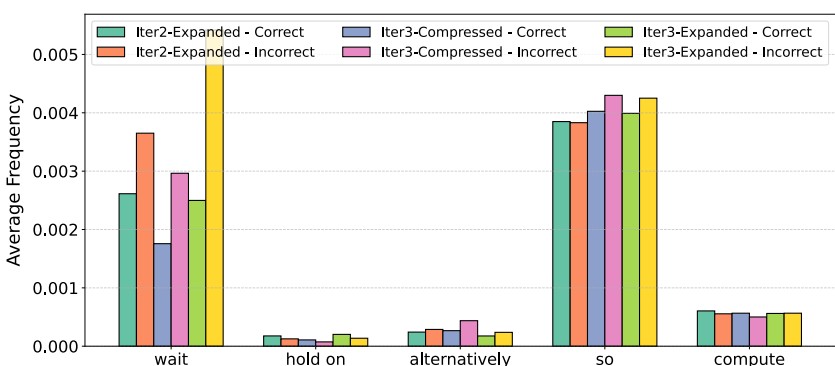

Figure 6: Representative token frequency before and after compression.

keeps pushing the Pareto frontier forward after each iteration, resulting in higher accuracy as well as greater token efficiency. Specifically, the accuracy of the compressed-length variant (SIRI-Iter1/2/3-Compressed) goes up across iterations: $33.3\% \rightarrow 38.1\% \rightarrow 40.4\%$, while the average response length continually goes down: $8065 \rightarrow 7266 \rightarrow 7093$.

To validate the advantage of our iterative compression-extension scheme over DeepScaleR's two-stage compression-then-extension approach, we compare SIRI with DeepScaleR-DAPO-16K under similar training times. As shown in Figure 5, SIRI reaches comparable performance on the AIME24 benchmark while largely outperforming it on the more challenging AIME25 benchmark with substantially fewer tokens due to the interleaved compression phases. These findings suggest that SIRI's iterative compression scheme effectively improves token efficiency and is better adapted to more demanding, reasoning-intensive tasks. A similar observation can also be drawn from the dynamics of the 7B model, as discussed in Appendix A.1.

**The iterative compression-expansion scheme mainly influences the model's backtracking and verification behavior**. We further analyze the change in the 1.5B model's behavior after compression and expansion. Specifically, we choose the model's responses for AIME24 problems at step 1280 (the finish of the second expansion stage), step 1600 (the finish of the third compression stage), and step 1920 (the finish of the third expansion stage) during the 640-cycle cosine schedule. We choose tokens that represent the model's backtrack-verification ("wait", "hold on"), alternative-seeking ("alternatively"), and general deduction behavior ("so", "compute"). As shown in Figure 6, the frequency of "wait" tokens that stand for backtracking and verification changes significantly during training, while others remain stable. In particular, the "wait" tokens are suppressed during compression and encouraged during expansion, and this trend is consistent for both correct and incorrect responses. Notably, the correct responses from the model at step 1280 and 1920 are almost identical, despite the latter having better performance. This shows that the interleaved compression phase indeed encourages the model to add more information under the same generation context.

**Entropy oscillation continually pushes model improvement**. In Appendix A.2, we also attempt to analyze SIRI's success from an entropy perspective. We observe that entropy decreases during

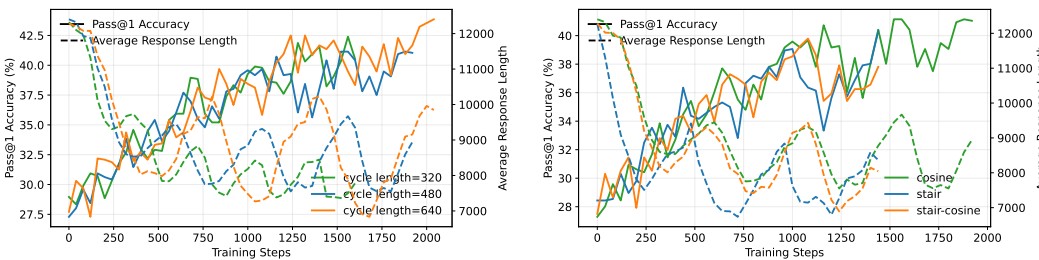

(a) Dynamics of cosine scheduler with different cycle lengths.

(b) Dynamics of different-shaped schedulers with a cycle length of 480.

Figure 7: Ablation studies on scheduler design.

compression and gradually increases during expansion, but remains stable within a bounded range rather than collapsing. Notably, performance gains often accompany rising entropy, allowing SIRI trained model to evolve through these oscillations.

## 5.3 Ablations on Scheduler Design

**Scheduler with a longer cycle performs best**. The design of the scheduler is the key to iterative improvement in each cycle. Figure 7a demonstrates the 1.5B model's performance of cosine scheduler with cycle lengths of 320, 480, and 640. During the compression stage, the 320-cycle and 480-cycle scheduler suffers from sharp performance degradation, while the 640-cycle scheduler reaches its response length minima with a mild drop in performance. In addition, the longer expansion phase of the 640-cycle scheduler ensures sustained and stable accuracy gains. As a whole, the 640-cycle scheduler leads to the largest length oscillation and highest compression ratio at the response length minima. This shows that a smoother compression phase is crucial for performance maintenance, while a longer expansion phase is the key to iterative accuracy improvement. This finding is in line with earlier work (Hou et al., 2025), where the authors argue that iterative length capping preserves performance, while direct length capping leads to sharp decline in response length, causing serious performance loss.

**Scheduler with different shapes has different advantages**. We show in Figure 7b the 1.5B model's performance of the stair, cosine, and stair-cosine schedulers, all with a cycle length of 480. We observe that the cosine scheduler mitigates performance loss during compression, while the stair scheduler maximizes performance gain during expansion. Specifically, in Figure 7b, the cosine scheduler maintains the model's accuracy around 0.39 when its response length falls from above 9000 to around 8000 from step 960 to step 1200. However, the performance drops further while the scheduler slowly increases the maximum generation length to 16K. In comparison, the direct 8K compression phase of the stair scheduler causes a sharp drop in the model's performance, but the subsequent full 16K expansion phase significantly boosts the model's response length and accuracy. For the stair scheduler, the extended 8K compression phase also fails to improve performance, while the extended 16K expansion phase brings additional gains. Again, this indicates that the compression phase should be smooth, while the expansion phase should be extended, relaxing its constraint on model's exploration behavior.

## 6 Conclusion

In this paper, we propose SIRI, a simple but effective approach to enhance the performance of LRMs while pruning repetitive reasoning traces. We apply expansion and compression of the token budget iteratively, encouraging exploration and consolidation in turn. Experiments show that SIRI boosts the model's performance and token efficiency consistently during each iteration. While this approach has provided extra gains, the upper performance threshold remains to be discovered and understood (e.g., limited by dataset size, algorithm efficiency, etc). Moreover, how SIRI can be applied in other tasks that require intensive reasoning, such as code generation, is also a promising direction. Looking forward, online RL post-training has been an ever-broadening avenue towards artificial general intelligence, and we hope this work can help to further scale up RL training.

## REPRODUCIBILITY STATEMENT

All datasets used for training and evaluation are open-sourced. Our training code is modified from the open-source framework VeRL, and we will release the modified parts upon publication. We will also release the trained model checkpoints along with the corresponding training logs.

## USE OF LARGE LANGUAGE MODELS (LLMS)

Large Language Models (LLMs) were used solely for polishing and enhancing the clarity of the manuscript. They did **not** contribute to research ideation, methodological design, experimental execution, data analysis, or any other substantive aspect of this work. All scientific content, results, and conclusions are entirely the responsibility of the authors.

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

# A ADDITIONAL EXPERIMENT RESULTS

## A.1 RESULTS ON DEEPSEEK-R1-DISTILL-QWEN-7B

Figure 8 demonstrates the training dynamics of SIRI and DAPO-DeepScaleR-16K for the DeepSeek-R1-Distill-Qwen-7B model. SIRI reaches comparable performance to DAPO-DeepScaleR-16K on AIME24 and outperforms DAPO-DeepScaleR-16K on AIME25, both with less tokens.

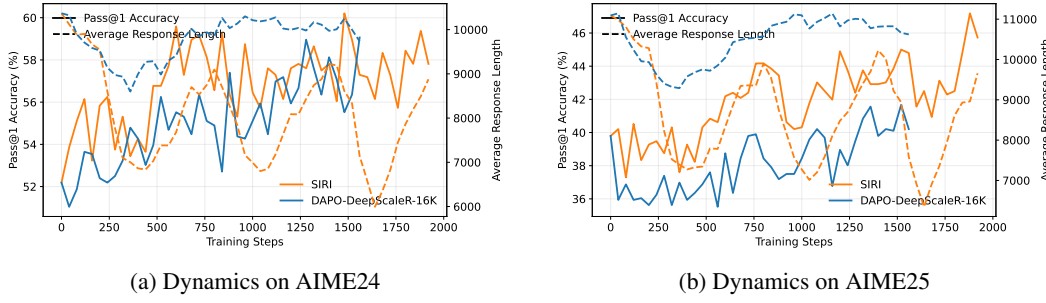

(a) Dynamics on AIME24                    (b) Dynamics on AIME25

Figure 8: The training dynamics comparison between SIRI and DAPO-DeepScaleR-16K. DAPO-DeepScaleR-16K transits from 8K to 16K at step 360.

## A.2 DETAILS ON TRAINING DYNAMICS: AN ENTROPY VIEW

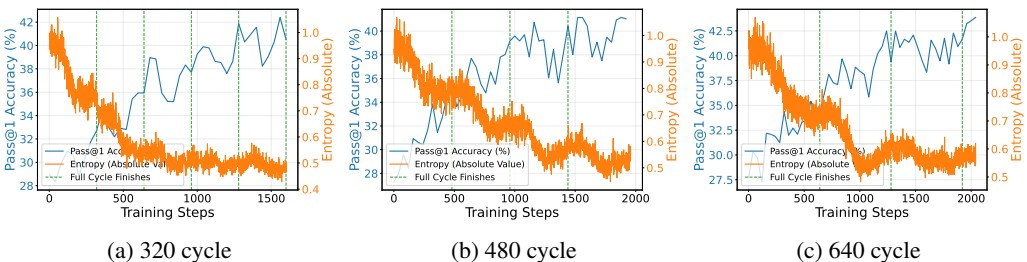

(a) 320 cycle                    (b) 480 cycle                    (c) 640 cycle

Figure 9: The entropy during training for cosine scheduler with different cycle length.

We additionally report the change of entropy during the cosine scheduler training in Figure 9. During the compression stage, the model's entropy decreases; During the expansion stage, its entropy slowly

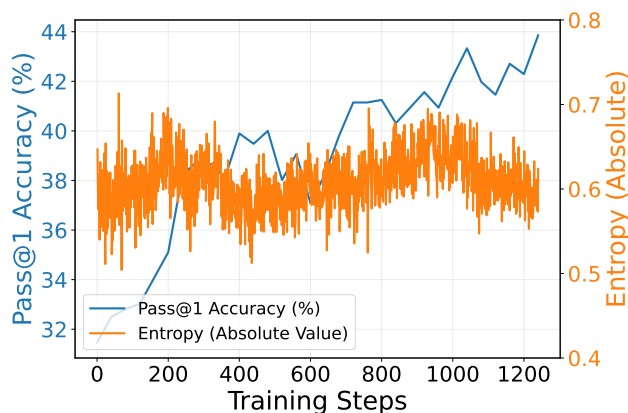

Figure 10: The entropy of DAPO-DeepScaleR-16K during 16K context training.

increases. However, we find that the model's entropy does not collapse. Instead, it tends to remain stable within a certain range as training proceeds.

Interestingly, for non-iterative models such as DAPO-DeepScaleR-16K, we notice similar trends, where the model's entropy periodically fluctuates. As shown in Figure 10, there is also roughly a cosine-shaped entropy curve during 16K context training of DAPO-DeepScaleR-16K. This shows that the periodic change in entropy is common for different training scheduler.

Moreover, for both training methods, we notice a increase in performance when entropy increases even as the response length pleataus for DAPO-DeepScaleR-16K after step 360. This implies the possibility of using entropy bonus or clipping even higher during the expansion stage to further enhance SIRI's performance.