# OpenReview forum: "SIRI: Scaling Iterative Reinforcement Learning with Interleaved Compression"
_ICLR.cc/2026/Conference — Submitted to ICLR 2026_

### Official Review · Reviewer_DQ7k · 2025-10-21

**Soundness:** 2
**Presentation:** 3
**Contribution:** 3
**Rating:** 2
**Confidence:** 3

**Summary:**

Oscillating response length helps the iterative on-policy RL mechanism such as GRPO.

**Strengths:**

* A simply designed length scheduler helps improve accuracy as iterations go by, as illustrated in Figures 1 and 4.
* If it is indeed true that accuracy increases when training with GRPO using responses generated through compression and expansion cycles, this represents a quite intriguing discovery.

**Weaknesses:**

* While the paper provides empirical validation (Fig 4, 5, 7), the core mechanism in Figure 2(b) is a hypothesis without theoretical justification.
* Unsure that the finding is statistically valid—Standard error or confidence interval is not reported in any experiments.
* The stylized dynamics in Figure 2(b) do not align well with the actual training curves in Figures 4 and 5. The paper appears to be drawing connections between the hypothesis and results that may not genuinely exist.
* The way of using the length scheduler is not clear. Did you include the term in the GRPO objective and add additional parameter to control its effect?

**Questions:**

* How can we determine the causal relationship between compression and efficiency gains and between expansion and exploration capability?
* Recently, I read the paper below [1], which showed that GRPO leads to longer responses. The authors removed two terms to make it more stable and control the response length. This might be relevant to your work on analyzing response length.

[1] Understanding R1-Zero-Like Training: A Critical Perspective, COLM 2025

---

### Official Review · Reviewer_y5FW · 2025-10-26

**Soundness:** 3
**Presentation:** 3
**Contribution:** 3
**Rating:** 6
**Confidence:** 3

**Summary:**

The paper introduces SIRI, a framework for training LRMs that enhances reasoning accuracy while reducing token usage by dynamically alternating between compression and expansion phases during RL. SIRI employs a scheduler to adjust maximum rollout lengths, forcing concise decision-making in compression phases to eliminate redundancy and enabling exploration in expansion phases for long-horizon planning. SIRI improves performance on math benchmarks, outperforming baselines such as DeepScaleR and AdaptThink by pushing the Pareto frontier of efficiency and accuracy.

**Strengths:**

- Paper is well presented, and easy to read
- Detailed empirical analyses, empirically validating the ideas of the proposed method
- Proposed method is easy to implement, and leads to a SOTA performance in terms of Pareto frontier.
- Compares multiple scheduler choices

**Weaknesses:**

- The main idea of the paper does not seem very novel, as it is extending DeepScaleR's compression-extension approach into the iterative training framework.
- Including a new scheduler adds a number of hyperparameters, e.g., scheduler type, L_max, L_min, T. Tuning such hyperparameters can significantly increase the computational burden.
- Only validated on DeepSeek R1 Distill Qwen models

**Questions:**

- Some recent papers on RLVR report that compressing the number of thinking tokens can degrade the model's general performance on different tasks other than the one being trained. Is there any severe degradation on model performance on general tasks when the compression is repeatedly done?

- While in the ablations it is discussed that scheduler with a longer cycles performs best, DeepScaleR is outperformed by proposed method where DeepScaleR can also be seen as a form of SIRI with extremely long cycles. What would be the potential reasons that the proposed method outperforms DeepScaleR? How would the performance trend be if we further increase the cycle lengths more than those experimented in the paper?

---

### Official Review · Reviewer_bwAV · 2025-10-30

**Soundness:** 3
**Presentation:** 2
**Contribution:** 2
**Rating:** 4
**Confidence:** 4

**Summary:**

This paper introduces SIRI, a training method for Large Reasoning Models. The core idea is to improve both reasoning accuracy and token efficiency through an iterative process. During training, the model alternates between a compression phase with a short generation limit and an expansion phase with a longer limit. The compression phase forces the model to generate more concise and dense reasoning, while the expansion phase allows for exploration and planning. The authors show empirically that this cyclical training pushes the model towards better performance and efficiency, outperforming existing methods on mathematical reasoning benchmarks.

**Strengths:**

1. The proposed method is simple, intuitive, and easy to implement. It introduces a novel training curriculum by dynamically adjusting the maximum generation length using a scheduler.

2. The paper provides strong empirical evidence to support its claims. The experiments are conducted on two different model sizes and evaluated on multiple standard mathematical reasoning benchmarks. The results in Table 1 clearly demonstrate that SIRI improves both accuracy and token efficiency, surpassing strong baselines.

3. This paper presents a thorough analysis of the method's behavior. It goes beyond final performance numbers and investigates the training dynamics, the effect of different schedulers, and the changes in the model's output patterns.

**Weaknesses:**

1. The conceptual novelty of the method appears to be an incremental extension of prior work. The idea of a compression phase followed by an expansion phase was already present in the DeepScaleR baseline. The primary contribution here is making this process iterative, which feels more like a refinement than a fundamentally new approach.

2. The comparison with baseline methods could be more robust. The results for several key baselines are incomplete in Table 1, which weakens the comparative claims. Additionally, the main competitive baseline, DAPO-DeepScaleR, is an author implementation, which raises questions about whether it was optimally tuned for a fair comparison.

3. The explanation for why the method works is based on indirect evidence. The analysis linking performance gains to the frequency of specific keywords like "wait" is correlational. It does not provide a deep, causal understanding of how the model's reasoning structure is fundamentally improved by the iterative training.

**Questions:**

1. The method can be viewed as a form of curriculum learning. How does the proposed iterative oscillation compare to a more standard curriculum that simply starts with a short generation length and gradually increases it over time without the compression cycles?

2. Regarding the DAPO-DeepScaleR baseline, could you provide more detail on its implementation? Specifically, was its single expansion phase given a comparable number of training steps to one of your expansion phases to ensure that the performance difference is due to the iterative nature of SIRI?

---

### Official Review · Reviewer_v74n · 2025-10-31

**Soundness:** 2
**Presentation:** 3
**Contribution:** 4
**Rating:** 2
**Confidence:** 4

**Summary:**

This paper proposes SIRI, a training regime for Large Reasoning Models (LRMs) that aims to solve the performance-efficiency trade-off1. The core idea is to iteratively alternate between "compressing" and "expanding" the model's reasoning budget by dynamically adjusting the maximum rollout length $L$ during reinforcement learning2. The authors use a "length scheduler," (e.g., a cosine scheduler) to manage this alternation.

**Strengths:**

The empirical results on benchmarks like AIME24 are strong—achieving high accuracy with reduced token counts

**Weaknesses:**

Lack of Theoretical Guarantee or Novel Algorithm:

Entirely Heuristic: The method is motivated by a hypothesis based on a visual inspection of a previous paper's training curve. There is no formal analysis of why this compression-expansion cycle is beneficial.

No New Algorithm: The paper does not introduce a new loss function or make any fundamental modifications to the RL algorithm. It builds on existing methods (GRPO/DAPO) and primarily proposes a novel scheduling strategy for one of the training hyperparameters—the maximum length $L$. The proposed schedule resembles a cosine learning rate schedule.

Weak Post-Hoc Analysis: The analysis provided is purely observational. For example, the entropy analysis in Appendix A.2 is interesting, but it merely observes that entropy oscillates. Worse, it even notes that the baseline model also shows periodic entropy fluctuations, which confuses the claim that this oscillation is the unique, driving factor behind SIRI's success .

**Questions:**

Please see the weakness

---

### Meta-Review · Area_Chair_s3Xd · 2026-01-07

**Summary:**

This paper proposes SIRI, an iterative RL training regime that alternates between compression and expansion phases by dynamically adjusting the maximum rollout length. Reviewers generally agree that the method is simple and easy to implement, and that the empirical results show improved accuracy–efficiency trade-offs on mathematical reasoning benchmarks. However, the overall reviewer consensus trends negative to borderline, with two clear rejects and the remaining reviews expressing only weak support. The main concerns focus on the paper’s limited conceptual novelty, largely heuristic nature, and lack of theoretical or causal justification. Several reviewers also question the rigor and completeness of the experimental validation, including missing statistical significance analysis and potentially insufficiently controlled baseline comparisons. Because there was no rebuttal, none of these concerns were addressed or clarified, leaving substantial doubts about the paper’s scientific contribution and reliability.

**Reviewer Concerns:**

There is no rebuttal.

**Reviewer Scores:**

Will not change, since there is no rebuttal from the authors.

---

### Decision · Program_Chairs · 2026-01-26

Reject